# Natural Killer Cell-Secreted IFN-γ and TNF-α Mediated Differentiation in Lung Stem-like Tumors, Leading to the Susceptibility of the Tumors to Chemotherapeutic Drugs

**DOI:** 10.3390/cells14020090

**Published:** 2025-01-10

**Authors:** Kawaljit Kaur, Angie Perez Celis, Anahid Jewett

**Affiliations:** 1Division of Oral Biology and Medicine, The Jane and Jerry Weintraub Center for Reconstructive Biotechnology, University of California School of Dentistry, 10833 Le Conte Ave, Los Angeles, CA 90095, USA; drkawalmann@g.ucla.edu (K.K.); acelis28@ucla.edu (A.P.C.); 2The Jonsson Comprehensive Cancer Center, University of California, Los Angeles, Los Angeles, CA 90095, USA

**Keywords:** NK cells, lung cancer, cancer stem cells (CSCs), cytotoxicity, chemo-drugs, IFN-γ, TNF-α, differentiation

## Abstract

We demonstrate that natural killer (NK) cells induce a higher cytotoxicity against lung cancer stem-like cells (hA549) compared to differentiated lung cancer cell lines (H292). The supernatants from split-anergized NK cells (IL-2 and anti-CD16 mAb-treated NK cells) induced differentiation in hA549. Differentiated lung cancer cell line (H292) and NK cells differentiated hA549 expressed reduced NK cell-mediated cytotoxicity but expressed higher sensitivity to chemotherapeutic drugs. This finding validated our previous reports demonstrating that the levels of tumor killing by NK cells and by chemotherapeutic drugs correlate directly and indirectly, respectively, with the stage and levels of tumor differentiation. We also demonstrate the role of IFN-γ and TNF-α in inducing tumor differentiation. NK cells’ supernatants or IFN-γ and TNF-α-induced tumor differentiation was blocked when we used antibodies against IFN-γ and TNF-α. Therefore, IFN-γ and TNF-α released from NK cells play a significant role in differentiating tumors, resulting in increased susceptibility of tumors to chemotherapeutic drugs. We also observed the different effects of MHC-class I antibodies in CSCs vs. differentiated tumors. Treatment with anti-MHC-class I decreased NK cell-mediated cytotoxicity in hA549 tumors, whereas it increased NK cell-mediated cytotoxicity when differentiated tumors were treated with antibodies against MHC-class I.

## 1. Introduction

Lung cancer is found to be the most commonly diagnosed cancer in both females and males [1,2,3], and it has an approximately 15–19% five-year survival rate [4,5]. Approximately 27% and 85% of all cancer-related and lung cancer-related deaths worldwide, respectively, occur due to non-small cell lung cancer (NSCLC) [6,7]. Adenocarcinoma, squamous cell carcinoma, and large-cell carcinomas were found to be the most common subtypes of NSCLC [8]. Risk factors for lung cancer are exposure to environmental pollution, asbestos, radon, and cigarette smoking [9,10].

Highly aggressive and progressively fatal lung adenocarcinoma was found to be resistant to conventional radiotherapy and chemotherapy [11]. The identification of cancer stem cells (CSCs) in lung cancer presents a therapeutic challenge [12]. CSCs are the tumor population that has the potential to self-renew and greatly contribute to lung cancer progression, tumor relapse, and drug resistance [13]. Lung CSCs are categorized by their surface expression of PD-L1, epithelial cell adhesion molecule (EpCAM), CD44, CD90, CXCR4 (CXC chemokine receptor type 4), and CD133 [14,15,16,17,18].

Natural killer (NK) cells, known for their anticancer function, are innate immune cells that comprise approximately 5 to 15% of the total peripheral blood mononuclear cell (PBMC) lymphocytes in humans [19]. We have previously shown that NK cells directly kill CSCs or inhibit tumor growth by differentiating tumors via secreted factors, especially IFN-γ and TNF-α [20]. The differentiation of CSCs leads to an increased MHC-class I surface expression in tumors, a reduced sensitivity to NK cell-mediated cytotoxicity against these differentiated CSCs, and an increased chemotherapy drug-induced killing against these differentiated CSCs [21,22]. Significant defects in NK cells’ function result in poor prognosis in cancer patients [23,24,25,26,27,28,29,30,31,32].

To understand the role of tumor differentiation in the chemotherapeutic sensitivity of lung cancer, we induced differentiation in hA549 lung cancer stem-like cell lines before they were exposed to chemotherapeutic drugs. hA549 is the most commonly used human NSCLC cell line for lung adenocarcinoma research [33,34]. The combinatorial treatment of IL-2 and anti-CD16 mAbs inhibits NK cell-mediated cytotoxicity but induces a significant release of IFN-γ and TNF-α in NK cells. These two cytokines play a crucial role in tumor differentiation. We demonstrate that lung cancer stem cells, upon differentiation, exhibit reduced sensitivity to NK cell-mediated cytotoxicity but are targets of chemo-drug-mediated tumor death.

## 2. Results

### 2.1. Enhanced Susceptibility of Chemotherapeutic Drugs Against Differentiated Tumors in Comparison to Their Stem-like Counterparts

Untreated, IL-2 treated, and IL-2 in combination with anti-CD16 mAb-treated NK cells were incubated overnight before they were used as effectors in the cytotoxic assay, as well as being used to determine the secretion levels of IFN-γ and TNF-α. As previously documented [20], IL-2-treated NK cells mediated the highest cytotoxicity (Figure 1A and Appendix A), whereas those treated with the combination of IL-2 and anti-CD16 mAbs demonstrated a decreased cytotoxic activity and increased secretion of IFN-γ and TNF-α (Appendix A). Our previous findings demonstrated the resistance to NK cell-mediated killing and increased sensitivity to chemotherapeutics in differentiated tumors in comparison to CSCs [22]. Here, we determined the NK cell-mediated cytotoxicity against hA549 (CSCs) and H292 (differentiated lung cancer cell lines) and found higher NK cell-mediated cytotoxicity against hA549 compared to H292 (Figure 1A). When the cell viability of untreated and chemotherapeutic drug-treated hA549 and H292 was assessed, both paclitaxel (Figure 1B) and CDDP (Figure 1C) induced higher cell death against H292 in comparison to hA549.

### 2.2. Enhanced Susceptibility of Chemotherapeutic Drugs Against NK Cell-Differentiated Tumors Compared to Stem-like Tumors

Our previous studies have demonstrated split-anergized NK cells secrete higher cytokines but mediate minimal cytotoxicity when treated with IL-2 + anti-CD16 mAbs and, that the supernatants from such NK cells induce the differentiation of stem cells [20]. CSCs were found to exhibit lower surface expression of CD54, MHC-class I, and PD-L1 (B7H1), and mediate higher sensitivity to NK cell-induced killing, while demonstrating increased resistance to chemo-drug-induced killing compared to differentiated tumors [20,22]. In this study, we treated hA549 with untreated NK cells’ or IL-2 combined with anti-CD16 mAb-treated NK cells’ supernatants. hA549 treated with IL-2 and ant-CD16 mAb-treated NK cells’ supernatants expressed a significant increase in CD54, MHC-class I, and B7H1 (Figure 2A), as well as a decreased sensitivity to NK cell-induced killing (Figure 2B) compared to untreated hA549 and untreated NK cells’ supernatant-treated hA549. The treatment of paclitaxel (Figure 2C) and CDDP (Figure 2D) induced significant hA549 cell death in IL-2 + anti-CD16 mAb-treated NK cells’ supernatant-differentiated hA549 compared to non-differentiated hA459 tumors.

### 2.3. Enhanced Susceptibility of rhIFN-γ- and rhTNF-α-Treated Tumors Against Chemotherapeutic Drugs 

It was shown earlier that IFN-γ and TNF-α play significant roles in tumor differentiation [20], therefore, we treated hA549 with rh-IFN-γ and rh-TNF-α to determine the IFN-γ- and TNF-α-induced differentiation in hA549 (Figure 3). We observed increased resistance to NK cell-mediated cytotoxicity in hA549 (Figure 3A) and increased sensitivity to paclitaxel- and CDDP-induced (Figure 3B and Figure 3C, respectively) cell death with rh-IFN-γ and rh-TNF-α treatments. An increased surface expression of CD54, MHC-class I, and B7H1 was also seen after rh-IFN-γ and rh-TNF-α treatment in hA549 (Appendix A).

### 2.4. Anti-IFN-γ and Anti-TNF-α Inhibited Both NK Cells’ Supernatant Treated, or rhIFN-γ and rhTNF-α Mediated Differentiation of hA549 Cells

When NK cells supernatant-treated hA549 cells were further treated with anti-IFN-γ or anti-TNF-α antibodies, we observed that the treatment with antibodies blocked NK cell-induced differentiation of hA549 cells, as indicated by no change in the surface markers (Figure 4A) and cytotoxicity assay profiles (Figure 4C). The greatest effect was seen with the combination of anti-IFN-γ and anti-TNF-α treatment. A minimal cell death was seen in hA549 when they were treated with anti-IFN-γ and/or anti-TNF-α (Figure 4B). To validate the role of IFN-γ and TNF-α further, we treated hA549 cells with rhIFN-γ and rhTNF-α in the absence and presence of antibodies against IFN-γ and TNF-α. We observed that rhIFN-γ, rhTNF-α, and the combination of rhIFN-γ with rhTNF-α induced the differentiation of hA549, which was blocked by anti-TNF-α, anti-IFN-γ, and anti-TNF-α + anti-IFN-γ (Figure 4D). Similar results were seen in SCAP cell lines (Appendix A). These results validated the role of IFN-γ and TNF-α in NK cell-induced tumor differentiation.

### 2.5. Anti-MHC-Class I Induces Different Effects in CSCs and Differentiated hA549 Cells

MHC-class I surface receptors were significantly increased in differentiated hA549 as compared to their stem-like counterpart. Untreated and IL-2-treated NK cells were used as effectors to induce cytotoxicity against hA549 and NK cell-differentiated hA549 using a chromium release assay. hA549 and NK cell-differentiated hA549 cells were treated with anti-MHC-class 1 (αPA2.6mAb) before they were incubated with effectors cells. When NK cell-mediated cytotoxicity levels were assessed, decreased NK cell-mediated cytotoxicity was observed in anti-MHC class I antibody-treated hA549 cells. In contrast, increased NK cell-mediated cytotoxicity was observed in NK cell-differentiated hA549 cells treated with antibodies against MHC-class I (Figure 5).

## 3. Discussion

Lung cancer was found as the leading and second-leading cause of cancer-related death in men and women, respectively [35]. The focus of this study is to delineate the underlying mechanisms by which lung cancer cells could undergo differentiation to be targeted by chemotherapy drugs. NK cells act as a powerful tool to induce differentiation in tumors [20]. We have previously described that IL-2 and anti-CD16 mAb-treated NK cells undergo ‘split anergy’ at the stage when NK cells express decreased cytotoxicity in the presence of a significant secretion of cytokines [20,36] (Appendix A). The cytokines, namely IFN-γ and TNF-α, released by the split-anergized NK cells promote the differentiation of tumors [20]. Differentiated tumors were found to exhibit lower CD44 and higher CD54, MHC-class I, and PD-L1 (B7H1) levels, and were found to exhibit a reduced sensitivity to the NK cell-induced killing, but were susceptible to chemo-drug-induced killing [20,22,37].

hA549 cells have features of CSCs, whereas H292 cell lines represent differentiated cell lines and they were analyzed to assess the key differences. The results indicated a significant correlation between the level of NK cell-induced killing and the stage of differentiation of tumors. Our previous studies using a 4-h chromium release assay have demonstrated that cancer stem-like tumors are excellent targets, whereas differentiated tumors are resistant to NK cell-mediated cytotoxicity [20,38]. Therefore, the purpose of the experiment in the current study is to compare different levels of NK cell-mediated cytotoxicity against lung cancer stem-like cells (hA549) and differentiated cell lines (H292). As mentioned earlier, treating NK cells with IL-2 and anti-CD16 mAbs induces split-anergy in NK cells—a stage where NK cells exhibit decreased cytotoxicity and increased cytokine secretion. We have observed the same finding in this study. The treatment of IL-2 and anti-CD16 antibodies in NK cells also resulted in the modulation of surface makers such as CD16, Nkp39, Nk-44, Nkp46, KIR2, KIR3, CD94, NKG2D, etc., as published in our previous studies [39,40]. Our previous work used oral cancer, pancreatic cancer, and glioblastoma to demonstrate that the supernatant of IL-2 and anti-CD16 antibody-treated NK cells, as well as IFN-γ and TNF-α, mediate the differentiation of cancer stem-like tumors [20]. Therefore, the purpose of treating lung cancer stem-like cells (hA549) with the supernatants of IL-2 and anti-CD16 antibody NK cells in the current study was to determine if IL-2 and anti-CD16 antibody-treated NK cells can mediate the differentiation of lung cancer stem-like cells as well. Similar to our previous observation in several other cancer types, the differentiated lung cancer cell line H292 was found to be more sensitive to chemotherapeutic drugs compared to hA549 [22]. Similar to differentiated lung cancer cell line H292, NK cell-induced differentiated hA549 tumors were found to exhibit reduced sensitivity to NK cell-induced killing but were highly susceptible to chemo-drug-induced killing compared to hA549 (Figure 2).

We have previously demonstrated that differentiated tumors express limited tumor growth or expansion [20]. Also, we have found a significant correlation between tumor killing by NK cells or the chemotherapeutic drugs and the differentiation stage of the tumors [20,22]. To determine whether recombinant human IFN-γ and TNF-α treatment of hA549 cells is capable of differentiating the hA549 tumor cells, we treated them with rhIFN-γ and rhTNF-α before we tested the NK cell-mediated cytotoxicity and chemo-drug induced killing of hA549 cells. We found that the treatment with rIFN-γ and rTNF-α of hA549 cells resulted in a decreased NK cell-mediated cytotoxicity and an increased chemo-drug-induced killing of hA549, as well as increasing the surface marker levels of B7H1, CD54, and MHC-class I in hA549 cells (Appendix A). These data indicate that differentiated lung cancer is also more susceptible to chemotherapeutic drugs when compared to CSC tumors; we have previously shown this finding in oral and pancreatic tumors [22].

To further validate the significance of NK cells’ secreted cytokines and/or rhIFN-γ and rhTNF-α in inducing differentiation in lung cancer, we treated tumors with antibodies against both IFN-γ and TNF-α to determine if we can block the NK cell- or rIFN-γ- and rTNF-α-induced differentiation of hA549. The antibodies against IFN-γ were found to induce a greater inhibition of hA549 tumor differentiation compared to antibodies against TNF-α. Significant blocking of hA549 differentiation was observed upon treatment with combined antibodies against TNF-α and IFN-γ, suggesting that split-anergized NK cells induce differentiation due to the synergistic function of TNF-α and IFN-γ. Upon treatment with antibodies against both TNF-α and IFN-γ in NK cells or rhIFN-γ- and rhTNF-α-differentiated hA549 cells, they regained NK cell-mediated cytotoxicity similar to hA549 CSCs. Similar effects were observed when SCAP cells were treated with either anergized NK supernatants or with recombinant IFN-γ and TNF-α, and induced differentiation was blocked when SCAP cells were treated further with antibodies against IFN-γ and TNF-α (Appendix A).

We observed that the levels of pJNK and pAKT were increased when hA549 cells were differentiated with NK cells or rIFN-γ and rTNF-α. The JNK and AKT pathways have been found to regulate cell death, and an increase in JNK and AKT correlates with increased cell survival [41,42]. No significant increase in STAT3 was observed when hA549 cells were treated with IFN-γ, TNF-α, and IFN-γ + rTNF-α, but STAT3 levels were increased when hA549 cells were treated with anergized NK cells’ supernatants, suggesting increased levels of STAT3 in lung tumors are specific to NK cell-induced differentiation. Interestingly, the treatment with antibodies against TNF-α and IFN-γ in hA549 cells pre-treated with anergized NK supernatants decreased STAT3, suggesting that TNF-α and IFN-γ play a role in the NK cell-mediated increase in STAT3. TNF-α and IFN-γ may be working synergistically with other NK cell-secreted cytokines, which awaits future investigation.

It was found that CSC tumors express lower MHC-class I, whereas well-differentiated tumors express increased MHC-class I levels on their surface [20,22,37]. In the current study, we observed the highest surface expression levels of MHC-class I in NK cells or IFN-γ- and TNF-α-induced differentiated hA549 cells. Increased surface expression levels of MHC-class I directly correlated with the NK cell-induced killing resistance of hA549. Similar results were seen in healthy stem cells such as SCAP and several other tumors, suggesting that this mechanism is not exclusive to lung tumors. To explore more on the significance or role of MHC-class I in inducing the resistance of tumors to NK cell-induced killing, we added antibodies against MHC-class I in untreated and NK supernatant-treated hA549 cells. Increased resistance to NK cell-mediated cytotoxicity was observed when untreated hA549 cells were treated with antibodies against MHC-class I, whereas the opposite effect was seen in NK cell-differentiated hA549 cells (Figure 5). Since hA549 cells express low levels of MHC-class I, it may be the case that adding anti-MHC-class I did not have a significant effect on the tumors [43,44]. Suppressed T cell function was seen with the treatment of antibodies against MHC [45]. On the other hand, when NK cell-differentiated hA549 cells were treated with anti-MHC-class I, we observed the exacerbation of lysis by NK cells. Differentiation in tumors increases MHC-class I levels and blocks NK cell-mediated cytotoxicity. When MHC-class I is blocked in differentiated tumors, the effect of an increase in NK cell-mediated cytotoxicity is highly elevated.

Overall, this study showed that NK cell-secreted cytokine IFN-γ and TNF-α induced differentiation and increased MHC-class I surface expression, which is one of the hallmarks of differentiation. Differentiated lung cancer was shown to exhibit reduced sensitivity to NK cell-induced killing but was sensitive to chemo-drug-induced killing.

## 4. Materials and Methods

### 4.1. Reagents and Antibodies for Cell Cultures and Flow Cytometry

Culture media for NK cells constitutes RPMI 1640 (Invitrogen by Life Technologies, CA, USA) combined with 10% fetal bovine serum (FBS) (Gemini Bio-Products, CA, USA). Human acinar adenocarcinoma cell line hA549 (Catalog# CCL-185) was purchased from ATCC (Manassas, VA, USA). hA549 cells were cultured in DMEM. Recombinant IL-2 used for NK cell treatment was obtained from NIH-BRB. Antibodies to CD16, recombinant human IFN-γ, recombinant human TNF-α, ELISA kits for IFN-γ, and flow cytometry antibodies were purchased from Biolegend (San Diego, CA, USA). Antibodies to MHC-class I (αPA2.6mAb), IFN-γ, and TNF-α were prepared in our laboratory, as described earlier. Human NK cell isolation kits were purchased from Stem Cell Technologies (Vancouver, BC, Canada). Propidium iodide (PI) and Chromium-51 were purchased from PeproTech (Cranbury, NJ, USA). Cisplatin and paclitaxel chemotherapeutic drugs were purchased from Ronald Reagan Pharmacy at UCLA.

### 4.2. Isolation of NK Cells from PBMCs

All of the procedures were approved by the UCLA Institutional Review Board (IRB), and written informed consent was obtained from healthy individuals. Ficoll-hypaque centrifugation methodology was used to isolate peripheral blood mononuclear cells (PBMCs) from peripheral blood. PBMCs were used to negatively select NK cells using the EasySep^®^ Human NK cell enrichment kit purchased from Stem Cell Technologies (Vancouver, BC, Canada). The purity of purified NK cells was assessed with CD16 antibodies using flow cytometric analysis, and the samples expressing greater than 95% purity were used for the experiments shown in this paper.

### 4.3. Lung Cancer Stem Cell Differentiation Using NK Cells’ Supernatant

Untreated or rh-IL-2 (1000 U/mL) and anti-CD16 mAb (3 µg/mL) treated NK cells were incubated overnight before the supernatant was harvested and was used to induce the differentiation of hA549 cells. IFN-γ levels in the NK cells’ supernatant were assessed using ELISA and the volume was determined based on the amount of IFN-γ required. hA549 cells were treated with an average total of 6000 pg over 4 days. hA549 (1 × 10^6^) cells were cultured overnight before unattached tumor cells were washed off and attached tumors were treated with NK cells’ supernatants, followed by daily NK cells’ supernatants for 4 days. On day 5, tumor cells were washed using 1 × PBS and were detached using trypsin.

### 4.4. Surface Markers and Cell Death Analysis

Surface markers were assessed after the cells were labeled with antibodies, as described previously [46]. Using flow cytometric (Beckman Coulter Epics XL cytometer (Brea, CA, USA) analysis, the percentage of viable cells was determined after the cells were stained with propidium iodide (PI) (100 μg/mL). Flow cytometric results were analyzed using FlowJo v10 software (Ashland, OR, USA).

### 4.5. Enzyme-Linked Immunosorbent Assays (ELISAs)

Cytokine levels were assessed using single ELISAs, as previously described [46]. A standard curve was generated by either two- or three-fold dilutions of recombinant cytokines provided by the manufacturer to analyze and obtain the cytokine concentration.

### 4.6. Four-Hour Chromium Release Cytotoxicity Assay

The ^51^Cr release cytotoxicity assay was performed as previously described [47]. Different ratios of NK cells and ^51^Cr-labeled lung cancer cell lines were incubated for four hours before the supernatants from each sample were harvested, and the released radioactivity was counted using a gamma counter. The percentage-specific cytotoxicity was calculated as follows:(1)$$\%\mathrm{cytotoxicity}=\frac{\mathrm{Experimental}\;\mathrm{cpm}-\mathrm{spontaneous}\;\mathrm{cpm}}{\mathrm{Total}\;\mathrm{cpm}-\mathrm{spontaneous}\;\mathrm{cpm}}$$

Lytic units (LU) 30/10^6^ were calculated by using the inverse of the number of NK cells needed to lyse 30% of ovarian cell lines × 100.

### 4.7. Statistical Analyses

Statistical analysis was performed using prism-10 software. The experiments with two groups were analyzed using an unpaired or paired two-tailed Student’s *t*-test. The experiments with more than two groups were analyzed using one-way ANOVA with a Bonferroni post-test. For cell cultures and other experiments, duplicate or triplicate samples were used. The following symbols represent the levels of statistical significance within each analysis: *** (*p* value < 0.001); ** (*p* value 0.001–0.01); * (*p* value 0.01–0.05).

## Figures and Tables

**Figure 1 cells-14-00090-f001:**
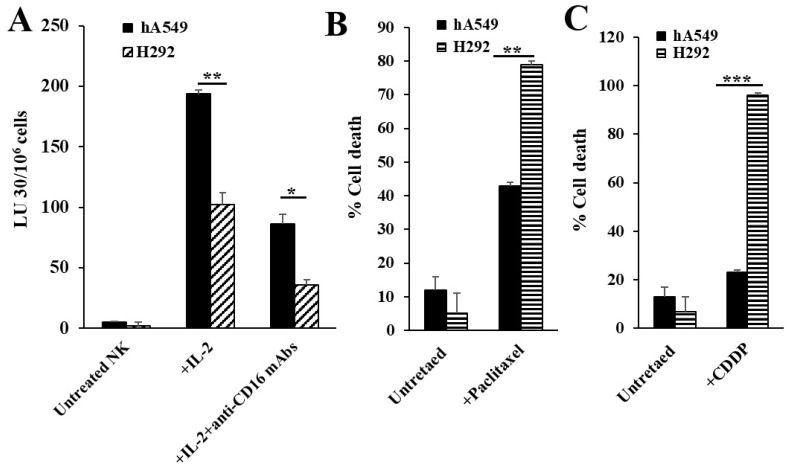
Different levels of killing of stem-like vs. differentiated tumors by NK cells and chemo-drugs. Untreated, IL-2 treated, and IL-2 + anti-CD16 mAb-treated NK cells (1 × 10^6^ cells/mL) of healthy individuals were incubated overnight before they were used as effector cells against ^51^Cr-labeled hA549 or H292 lung cancer cell lines at different NK-to-tumor (Effector: Target) ratios using a 4-h ^51^Cr release assay. The lytic unit (LU) 30/10^6^ cells were calculated based on an inverse number of NK cells being needed to kill 30% of the target cells × 100 (**A**). hA549 and H292 lung cancer cell lines were treated with paclitaxel (40 µg/mL) (**B**) and cisplatin (CDDP) (40 µg/mL) (**C**) overnight before the tumors were stained with propidium iodide (PI); percent cell viability was determined using a flow cytometer (**B**,**C**). *** (*p* value < 0.001); ** (*p* value 0.001–0.01); * (*p* value 0.01–0.05).

**Figure 2 cells-14-00090-f002:**
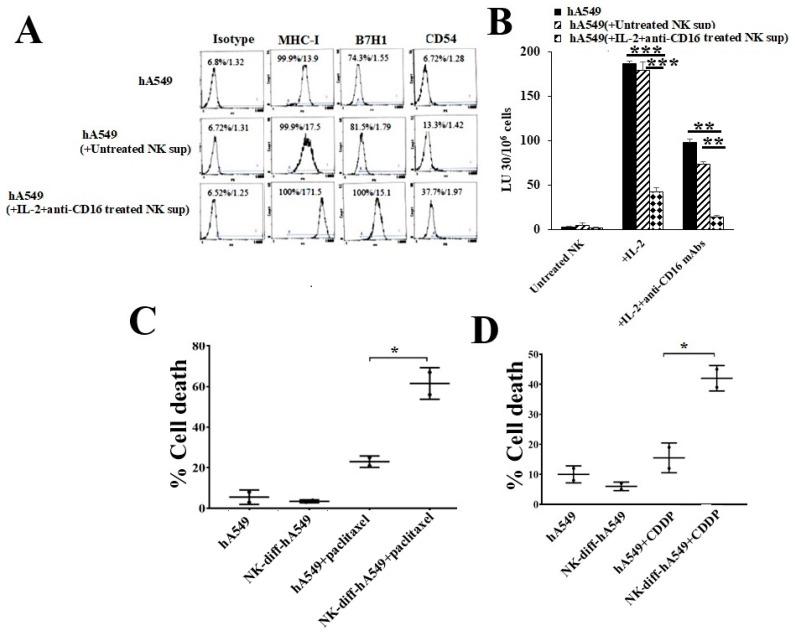
IL-2 and anti-CD16 mAb-treated NK cells’ supernatants induced differentiation in hA549 tumors. As described in the Materials and Methods, hA549 tumors were differentiated using untreated and IL-2 + anti-CD16 mAb-treated NK cells’ supernatants. hA549 tumors were analyzed for IgG2, MHC-class I, B7H1 (PD-L1), and CD54 surface expression using flow cytometer (**A**). Untreated, IL-2 treated and IL-2 + anti-CD16 mAb-treated NK cells (1 × 10^6^ cells/mL) from healthy individuals were incubated overnight before they were used as effector cells against ^51^Cr-labeled CSCs and differentiated hA549 at different NK-to-tumor (Effector: Target) ratios using a 4-h ^51^Cr release assay. The lytic unit (LU) 30/10^6^ cells were calculated based on an inverse number of NK cells needed to kill 30% of the target cells × 100 (**B**). hA549 and NK cell-differentiated hA549 were treated with paclitaxel (40 µg/mL) (**C**) or with cisplatin (CDDP) (40 µg/mL) (**D**) overnight before the tumors were stained with propidium iodide (PI), and percent cell viability was determined using flow cytometric analysis (**C**,**D**). *** (*p* value < 0.001); ** (*p* value 0.001–0.01); * (*p* value 0.01–0.05).

**Figure 3 cells-14-00090-f003:**
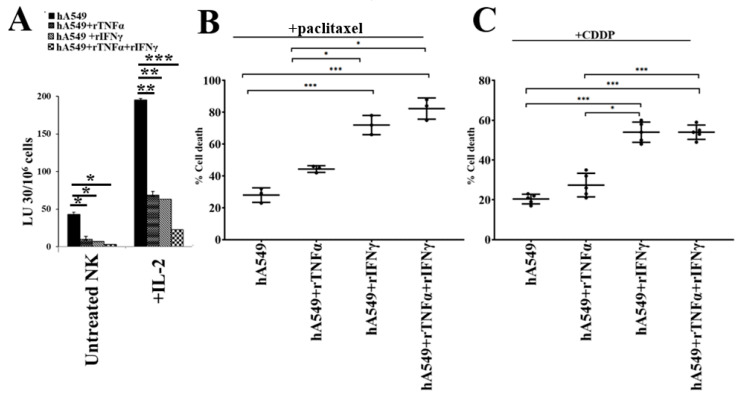
Treatment with IFN-γ and TNF-α induced differentiation in hA549 tumors. hA549 cells treated with rhTNF-α (20 ng/mL), rhIFN-γ (200 U/mL), and rhTNF-α (20 ng/mL) + rhIFN-γ (200 U/mL) were incubated overnight before they were used as targets for untreated and IL-2-treated NK cells in a 4-h ^51^Cr release assay. The lytic unit (LU) 30/10^6^ cells were calculated based on an inverse number of NK cells needed to kill 30% of the target cells × 100 (**A**). Untreated and rhTNF-α (20 ng/mL), rhIFN-γ (200 U/mL), and rhTNF-α (20 ng/mL) + rhIFN-γ (200 U/mL)-treated hA549 cells were treated with paclitaxel (40 µg/mL) (**B**) and cisplatin (CDDP) (40 µg/mL) (**C**) overnight before they were stained with propidium iodide (PI), amd percent cell viability was determined using flow cytometric analysis (**B**,**C**). *** (*p* value < 0.001); ** (*p* value 0.001–0.01); * (*p* value 0.01–0.05).

**Figure 4 cells-14-00090-f004:**
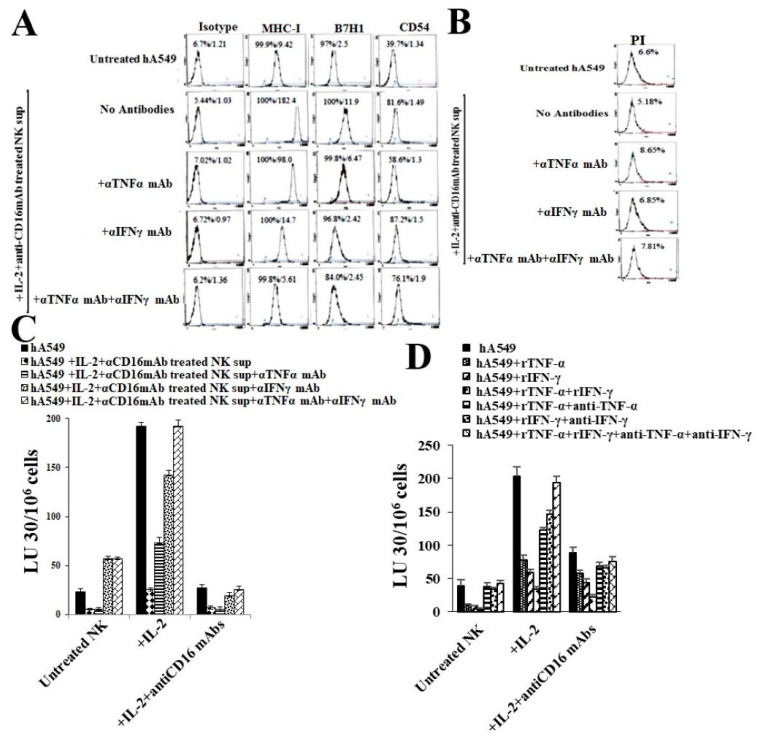
Treatments with antibodies against IFN-γ and TNF-α blocked NK cells’ supernatant-induced, as well as the IFN-γ- and TNF-α-induced, differentiation of hA549 cells. As described in the Materials and Methods, hA549 tumors were treated with supernatants from IL-2 + anti-CD16 mAb-treated NK cells alone or in combination with rhTNF-α (20 ng/mL), rhIFN-γ (200 U/mL), and αTNFα mAbs (1:100) + αIFNγ mAbs (1:100) for six days. On day 6, tumors were analyzed for surface marker levels of IgG2, CD54, MHC-class I, B7H1 (PD-L1), and CD54 using flow cytometric analysis (**A**). hA549 cells were treated as described in (**A**) and were stained with propidium iodide (PI) and percent cell viability was determined using flow cytometer (**B**). hA549 cells were treated as described in (**A**). Untreated, IL-2 (1000 U/mL)-treated, and IL-2 + anti-CD16 mAb (3 μg/mL)-treated NK cells were incubated overnight before they were used as effectors against hA549 cells in a 4-h ^51^Cr release assay. The lytic unit (LU) 30/10^6^ cells were calculated based on an inverse number of NK cells needed to kill 30% of the target cells × 100 (**C**). hA549 tumors were treated with rhTNF-α (20 ng/mL), rhIFN-γ (200 U/mL), rhTNF-α (20 ng/mL) + rhIFN-γ (200 U/mL), in the presence and absence of αIFN-γ mAbs (1:100), and αTNFα mAbs (1:100) for six days, and were used as targets for untreated, IL-2 treated, and IL-2 + anti-CD16 mAb-treated NK cells in a 4-h ^51^Cr release assay. The lytic unit (LU) 30/10^6^ cells were calculated based on an inverse number of NK cells needed to kill 30% of the target cells × 100 (**D**).

**Figure 5 cells-14-00090-f005:**
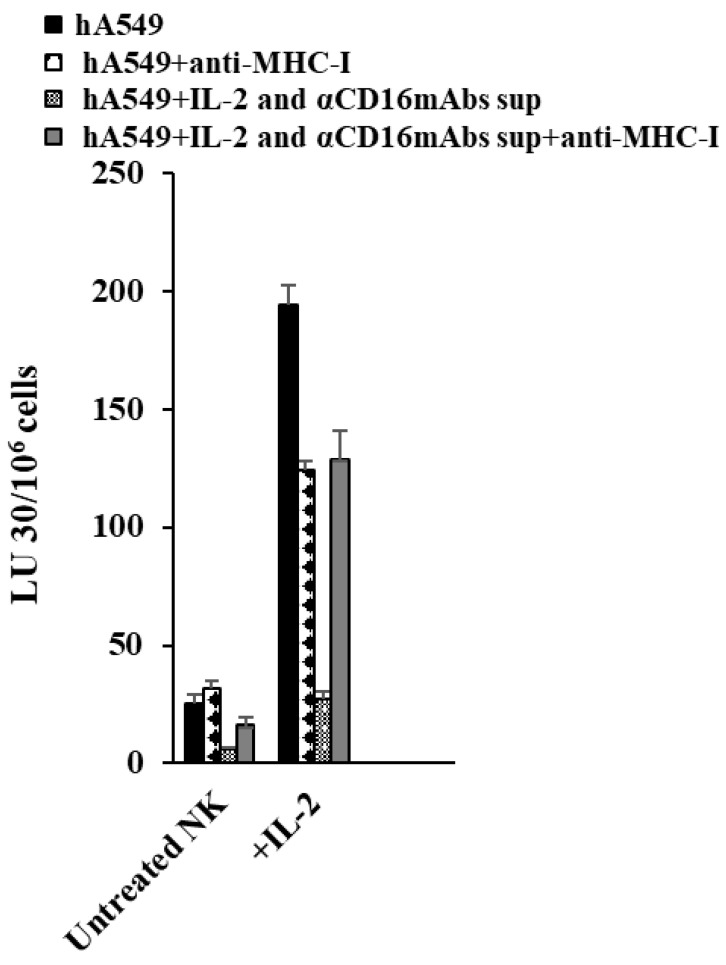
Treatment of hA549 with anti-MHC-class I (αPA2.6mAb, 1:100) decreased the sensitivity of hA549 CSCs and increased the sensitivity of differentiated hA549 cells, respectively, to NK cell-induced tumor killing. As described in the Materials and Methods, IL-2 and anti-CD16 mAb-treated NK cells’ supernatant was used to differentiate hA549 cells. Untreated and IL-2 (1000 U/mL)-treated NK cells were incubated overnight before they were used as effectors against hA549 cells in a 4-h ^51^Cr release assay. The lytic unit (LU) 30/10^6^ cells were calculated based on an inverse number of NK cells needed to kill 30% of the target cells × 100.

## Data Availability

Data generated or analyzed during the study are included in this submitted article.

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
