# Peer review of "Natural Killer Cell-Secreted IFN-γ and TNF-α Mediated Differentiation in Lung Stem-like Tumors, Leading to the Susceptibility of the Tumors to Chemotherapeutic Drugs"

_cells, 2025, doi:10.3390/cells14020090_

Round 1

Reviewer 1 Report

Comments and Suggestions for Authors

1. Supplementary files are attached to the main article. They should be added separately. What is more there practically the same as those in the manuscript.

2. Paragraph 2.1 First sentence has to many “or”. It should be rewritten.
As well as “As shown previously in several publications from our laboratory”. In addition, appropriate citations should be added.

3. Paragraph 2.2 When describing results “higher increase” should not be used. Instead, it should be underlined where statistically significant differences were determined.

Figure 2A has very low quality.

4. Paragraph 2.4 I don’t really understand the first sentence. Is the second sentence continuation?
Instead of “no” word “none” should be used too.
Again to many “or” were used in that paragraph.
Figure 4 has low quality.

5. Paragraph 2.5. Instead of “seen” another word should be used. This paragraph should also be described in more detailed way.

6. Paragraph 3 “We have previously introduced the term ‘split anergy’ in NK cells. When treated with a combination of IL-2 and anti-CD16 mAbs, NK cells undergo a split-anergized state that demonstrates reduced NK cell cytotoxicity in the presence of significant secretion of cytokines” – these sentences should be put in one.

7. Paragraph 4.1 “Antibodies to CD16, recombinant IFN-γ, and recombinant TNF-α were from Biolegend (San Diego, CA)” – I understand that these antibodies were purchased or obtained from Biolegend.

“Antibodies to MHC-class I, IFN-γ and TNF-α were prepared in our laboratory” – there is a need for explanation of how these antibodies were “prepared” or a simple citation.

Catalogue numbers and clones of antibodies should also be provided.

7. Paragraph 4.4. Propidium Iodine is used mainly for assessing cell viability and exclusion of non-viable cells. It is a bit of an exaggeration describing it as a cell death since you can’t tell how many early and late apoptotic cells you had.

Overall this article is interesting, the aim of the study is clearly described and well argued, methods have been appropriately selected.

Author Response

  1. Supplementary files are attached to the main article. They should be added separately. What is more there practically the same as those in the manuscript.

Response: We have now separated and re-uploaded the correct supplementary file.

  1. Paragraph 2.1 First sentence has to many “or”. It should be rewritten.
    As well as “As shown previously in several publications from our laboratory”. In addition, appropriate citations should be added.

Response: We have revised the information as suggested by the reviewer. Here is the revised section:

NK cells were left untreated, treated with IL-2 alone, and treated with a combination of IL-2 and anti-CD16 mAbs for 18-24 hours before they were used as effectors in the cytotoxic assay as well as to determine the secretion levels of IFN-γ and TNF-α. As previously documented 1, IL-2-treated NK cells mediated the highest cytotoxicity (Figs. 1A and S1A), whereas those treated with the combination of IL-2 and anti-CD16m Abs demonstrated decreased cytotoxic activity in the presence of increased secretion levels of IFN-γ and TNF-α (Fig. S2). Our previous findings have demonstrated that differentiated tumors were resistant to NK cell-mediated killing but were more sensitive to chemotherapeutics in comparison to CSCs 2. Here, we determined the NK cell-mediated cytotoxicity against hA549 (CSCs) and H292 (differentiated lung cancer cell lines) and found higher NK cell-mediated cytotoxicity against hA549 compared to H292 (Fig. 1A). When cell viability of untreated and chemotherapeutic drug-treated hA549 and H292 was assessed, both paclitaxel (Fig. 1B) and CDDP (Fig. 1C) induced higher cell death against H292 in comparison to hA549.

  1. Paragraph 2.2 When describing results “higher increase” should not be used. Instead, it should be underlined where statistically significant differences were determined.

Figure 2A has very low quality.

Response: We have now improved the quality of the figure 2.

We have revised the information as suggested by the reviewer. Here is the revised section:

In this study, we treated hA549 with untreated NK cells as well as IL-2 and anti-CD16 mAbs-treated NK cells’ supernatants. hA549 treated with IL-2 and ant-CD16 mAbs treated NK cells’ supernatants expressed a significant increase of MHC-class I, B7H1, and CD54 (Fig. 2A) and increased resistance to NK cell-mediated cytotoxicity (Fig. 2B) in comparison to untreated hA549 and those treated with untreated NK cells’ supernatant. The treatment of paclitaxel (Fig. 2C) and CDDP (Fig. 2D) induced significant hA549 cell death in IL-2 and anti-CD16 mAbs treated NK cells’ supernatants differentiated hA549 compared to non-differentiated hA459 tumors.

  1. Paragraph 2.4 I don’t really understand the first sentence. Is the second sentence continuation?
    Instead of “no” word “none” should be used too.
    Again to many “or” were used in that paragraph.
    Figure 4 has low quality.

Response: We have now revised the paragraph, here is the revised section”

When NK cells supernatant treated hA549 cells were further treated with anti-IFN-γ or anti-TNF-α antibodies, we observed that treatment of antibodies blocked NK cells induced differentiation in hA549 cells as indicated by no change in the surface markers (Fig. 4A) and cytotoxicity assay profiles (Fig. 4C), the highest effect was seen with the combination of anti-IFN-γ and anti-TNF-α treatment. A minimal level of cell death was seen in hA549 when they were treated with anti-IFN-γ and/or anti-TNF-α (Fig. 4B). To validate the role of IFN-γ and TNF-α further, we treated hA549 cells with rhIFN-γ and rhTNF-α in the absence and presence of anti-IFN-γ and anti-TNF-α antibodies. We observed that rIFN-γ, rTNF-α and the combination of rIFN-γ with rTNF-α induced differentiation of hA549 which was blocked by anti-TNF-α, anti-IFN-γ and anti-TNF-α+anti-IFN-γ (Fig. 4D). Similar results were seen in SCAP cell lines (Figs. S4 and S5). These results validated the role of IFN-γ and TNF-α in NK cells induced differentiation of CSCs.

Also,  we replaced Figure 4 with the high-quality figure.

5. Paragraph 2.5. Instead of “seen” another word should be used. This paragraph should also be described in more detailed way.

Response: We have now revised the paragraph, here is the revised section

MHC-class I is one of the surface receptors that is significantly increased in differentiated hA549 compared to their stem-like counterpart. Untreated and IL-2-treated NK cells were used as effectors to induce cytotoxicity against hA549 and NK cell-differentiated hA549 using chromium release assay. hA549 and NK cell-differentiated hA549 were treated with anti-MHC-class 1 (αPA2.6mAb) before they were incubated with effector cells. When NK cell-mediated cytotoxicity was assessed, decreased NK cell-mediated cytotoxicity was observed in anti-MHC class I antibody-treated hA549 cells. In contrast, increased NK cell-mediated cytotoxicity was observed in NK cell-differentiated hA549 treated with antibodies against MHC-class I (Fig. 5).

  1. Paragraph 3 “We have previously introduced the term ‘split anergy’ in NK cells. When treated with a combination of IL-2 and anti-CD16 mAbs, NK cells undergo a split-anergized state that demonstrates reduced NK cell cytotoxicity in the presence of significant secretion of cytokines” – these sentences should be put in one.

Response: We have now revised the sentence, here is the revised section

We have previously found that IL-2 and anti-CD16 mAbs treated NK cells undergo ‘split anergy’, a stage when NK cells express decreased cytotoxicity in the presence of significant secretion of cytokines 3-5

  1. Paragraph 4.1 “Antibodies to CD16, recombinant IFN-γ, and recombinant TNF-α were from Biolegend (San Diego, CA)” – I understand that these antibodies were purchased or obtained from Biolegend.

“Antibodies to MHC-class I, IFN-γ and TNF-α were prepared in our laboratory” – there is a need for explanation of how these antibodies were “prepared” or a simple citation.

Catalogue numbers and clones of antibodies should also be provided.

Response: We have now corrected and added missing information as suggested by the reviewer, here is the revised section:

Antibodies to CD16, recombinant IFN-γ, and recombinant TNF-α were purchased from Biolegend (San Diego, CA). Antibodies to MHC-class I (αPA2.6mAb), IFN-γ and TNF-α were prepared in our laboratory as described earlier 1.

  1. Paragraph 4.4. Propidium Iodine is used mainly for assessing cell viability and exclusion of non-viable cells. It is a bit of an exaggeration describing it as a cell death since you can’t tell how many early and late apoptotic cells you had.

Response: We have now revised section as: The percentage of viable cells was determined by propidium iodine (PI) (100 μg/ml) staining using flow cytometric analysis.

Overall this article is interesting, the aim of the study is clearly described and well argued, methods have been appropriately selected.

Response: We appreciate the kind words of the reviewer.

1             Bui, V. T. et al. Augmented IFN-γ and TNF-α Induced by Probiotic Bacteria in NK Cells Mediate Differentiation of Stem-Like Tumors Leading to Inhibition of Tumor Growth and Reduction in Inflammatory Cytokine Release; Regulation by IL-10. Frontiers in Immunology 6, doi:10.3389/fimmu.2015.00576 (2015).

2             Kozlowska, A. K. et al. Differentiation by NK cells is a prerequisite for effective targeting of cancer stem cells/poorly differentiated tumors by chemopreventive and chemotherapeutic drugs. J Cancer 8, 537-554, doi:10.7150/jca.15989 (2017).

3             Magister, S. et al. Regulation of cathepsins S and L by cystatin F during maturation of dendritic cells. Eur J Cell Biol 91, 391-401, doi:10.1016/j.ejcb.2012.01.001 (2012).

4             Tseng, H. C., Bui, V., Man, Y. G., Cacalano, N. & Jewett, A. Induction of Split Anergy Conditions Natural Killer Cells to Promote Differentiation of Stem Cells through Cell-Cell Contact and Secreted Factors. Front Immunol 5, 269, doi:10.3389/fimmu.2014.00269 (2014).

5             Tseng, H. C., Cacalano, N. & Jewett, A. Split anergized Natural Killer cells halt inflammation by inducing stem cell differentiation, resistance to NK cell cytotoxicity and prevention of cytokine and chemokine secretion. Oncotarget 6, 8947-8959, doi:10.18632/oncotarget.3250 (2015).

Reviewer 2 Report

Comments and Suggestions for Authors

This manuscript provides interesting findings about the correlation between the status of NK cells and the susceptibility of tumors to cancer drugs.

The are several major and minor points that need to be addressed before to make this manuscript suitable for publication:

MAJOR POINTS:

1) Authors treated NK cells with IL-2 for 18-24h before to use them as effectors in cytotoxic assay. What is the explanation of choosing 18-24h? Did the author try to treat NK cells for 48h or 72h to see if there is any difference in the outcome compared to 18-24h treatment?

2) In figure 1 authors reported the 4h killing assay, which is usually used for ADCC assay. Blocking the CD16 on NK cells with anti-CD16 mAbs affects the ADCC activity of NK cells, but no antibody able to mediate the ADCC was included in the assay. NK cells can still kill directly cancer cells after 16-18h from incubation with their target. Usually to measure NK direct killing, is more appropriate to use a 16-18h killing assay, because 4h killing assay will show only partial killing from NK cells. Since authors did not use a mAb able to mediate ADCC what is the meaning of 4h killing assay? Why author did not show the difference in killing between 4h and 16-18h killing assay?

3) Treatment of NK cells with IL-2 and anti-CD16 antibody not only modulates cytokine production by NK cells, but also modulate the expression of activation and cytotoxic markers on NK cells. This modulation affects cytotoxic activity of NK cells. Why did the author not evaluate the modulation of NK markers between treated and untreated NK cells by flow cytometry?

MINOR POINTS:

1) Abstract, line 3: replace IL- with IL-2

2) Introduction: include a reference at the end of the sentence beginning with NK cells know for their anticancer function are innate immune cells, that comprise 5 to 20% of PBMCs in humans.

Author Response

This manuscript provides interesting findings about the correlation between the status of NK cells and the susceptibility of tumors to cancer drugs.

The are several major and minor points that need to be addressed before to make this manuscript suitable for publication:

MAJOR POINTS:

  • Authors treated NK cells with IL-2 for 18-24h before to use them as effectors in cytotoxic assay. What is the explanation of choosing 18-24h? Did the author try to treat NK cells for 48h or 72h to see if there is any difference in the outcome compared to 18-24h treatment?

Response: We have previously analyzed the function and surface markers of IL-2-activated NK cells on days 1, 3, and 6 of activation. We have observed downmodulation of CD16 and other activation markers, reduced cell viability, and decreased cytotoxicity and cytokine secretion after 48 hours compared to 24 hours so we have selected 18-24 hours treatment for cytotoxic NK cells.

  • In figure 1 authors reported the 4h killing assay, which is usually used for ADCC assay. Blocking the CD16 on NK cells with anti-CD16 mAbs affects the ADCC activity of NK cells, but no antibody able to mediate the ADCC was included in the assay. NK cells can still kill directly cancer cells after 16-18h from incubation with their target. Usually to measure NK direct killing, is more appropriate to use a 16-18h killing assay, because 4h killing assay will show only partial killing from NK cells. Since authors did not use a mAb able to mediate ADCC what is the meaning of 4h killing assay? Why author did not show the difference in killing between 4h and 16-18h killing assay?

Response: We have used the standard 4-hour chromium assay to assess direct NK cell-mediated cytotoxicity against lung cancer cell lines and not the ADCC. Our previous studies using 4-hour chromium release assay have demonstrated that cancer stem-like tumors are excellent targets, whereas differentiated tumors are resistant to NK cell-mediated cytotoxicity 1-5. So, the purpose of this experiment in the current study is to show the different levels of NK cell-mediated cytotoxicity against lung cancer stem-like cells hA549 and differentiated cell lines H292. Moreover, in 16-18 hours of killing since we start losing the viability of NK cells the results are not as accurate as can be seen in 4 hour assay.

  • Treatment of NK cells with IL-2 and anti-CD16 antibody not only modulates cytokine production by NK cells, but also modulate the expression of activation and cytotoxic markers on NK cells. This modulation affects cytotoxic activity of NK cells. Why did the author not evaluate the modulation of NK markers between treated and untreated NK cells by flow cytometry?

Response: We agree with the reviewer that treatment of IL-2 and anti-CD16 antibodies results in increased cytokine secretion as well as modulation of surface makers such as CD16, Nkp39, Nk-44, Nkp46, KIR2, KIR3, CD94, and NKG2D, etc. These findings related to the modulation of surface markers were published in our previous studies 5-7. Our previous work used oral cancer, pancreatic cancer, and glioblastoma to demonstrate that supernatant of IL-2 and anti-CD16 antibodies treated NK cells as well as IFN-γ and TNF-α mediate differentiation of cancer stem-like tumors 2,8. Therefore, the purpose of treating lung cancer stem-like cells hA549 with the supernatants of IL-2 and anti-CD16 antibodies NK cells in the current study was to determine if IL-2 and anti-CD16 antibodies treated NK cells can mediate differentiation of lung cancer stem-like cells as well.

MINOR POINTS:

  • Abstract, line 3: replace IL- with IL-2

Response: Done

2) Introduction: include a reference at the end of the sentence beginning with NK cells know for their anticancer function are innate immune cells, that comprise 5 to 20% of PBMCs in humans.

Response: Done

1             Jewett, A., Man, Y.-G. & Tseng, H.-C. Dual Functions of Natural Killer Cells in Selection and Differentiation of Stem Cells; Role in Regulation of Inflammation and Regeneration of Tissues. Journal of Cancer 4, 12-24, doi:10.7150/jca.5519 (2013).

2             Tseng, H. C., Bui, V., Man, Y. G., Cacalano, N. & Jewett, A. Induction of Split Anergy Conditions Natural Killer Cells to Promote Differentiation of Stem Cells through Cell-Cell Contact and Secreted Factors. Frontiers in immunology 5, 269, doi:10.3389/fimmu.2014.00269 (2014).

3             Jewett, A. & Bonavida, B. Target-induced inactivation and cell death by apoptosis in a subset of human NK cells. Journal of Immunology 156, 907-915 (1996).

4             Tseng, H.-C. et al. Increased Lysis of Stem Cells but Not Their Differentiated Cells by Natural Killer Cells; De-Differentiation or Reprogramming Activates NK Cells. Plos One 5, doi:10.1371/journal.pone.0011590 (2010).

5             Kaur, K. et al. Novel Strategy to Expand Super-Charged NK Cells with Significant Potential to Lyse and Differentiate Cancer Stem Cells: Differences in NK Expansion and Function between Healthy and Cancer Patients. Frontiers in Immunology 8, doi:10.3389/fimmu.2017.00297 (2017).

6             Tseng, H. C. et al. Bisphosphonate-induced differential modulation of immune cell function in gingiva and bone marrow in vivo: role in osteoclast-mediated NK cell activation. Oncotarget 6, 20002-20025, doi:10.18632/oncotarget.4755 (2015).

7             Kaur, K. et al. Sequential therapy with supercharged NK cells with either chemotherapy drug cisplatin or anti-PD-1 antibody decreases the tumor size and significantly enhances the NK function in Hu-BLT mice. Frontiers in Immunology 14, doi:10.3389/fimmu.2023.1132807 (2023).

8             Bui, V. T. et al. Augmented IFN-γ and TNF-α Induced by Probiotic Bacteria in NK Cells Mediate Differentiation of Stem-Like Tumors Leading to Inhibition of Tumor Growth and Reduction in Inflammatory Cytokine Release; Regulation by IL-10. Frontiers in immunology 6, doi:10.3389/fimmu.2015.00576 (2015).

Round 2

Reviewer 2 Report

Comments and Suggestions for Authors

I thank authors for their responses.

Please, include in the discussion information provided in the response of question 2 and 3

Author Response

We appreciate the reviewer's time and efforts and have added a suggested section in the discussion as below.

hA549 has features of CSCs whereas H292 cell lines represented differentiated cell lines and they were analyzed to assess the key differences. The results indicated a significant correlation between the level of NK cell-induced killing and the stage of differentiation of tumors. Our previous studies using 4-hour chromium release assay have demonstrated that cancer stem-like tumors are excellent targets, whereas differentiated tumors are resistant to NK cell-mediated cytotoxicity 35,43-46. So, the purpose of this experiment in the current study is to compare different levels of NK cell-mediated cytotoxicity against lung cancer stem-like cells hA549 and differentiated cell lines H292. As mentioned earlier, treating NK cells with IL-2 and anti-CD16 mAbs induces split anergy in NK cells, a stage of NK cells exhibiting decreased cytotoxicity and increased cytokine secretion. We have observed the same finding in this study. The treatment of IL-2 and anti-CD16 antibodies in NK cells also resulted in modulation of surface makers such as CD16, Nkp39, Nk-44, Nkp46, KIR2, KIR3, CD94, and NKG2D, etc. as published in our previous studies 46-48. Our previous work used oral cancer, pancreatic cancer, and glioblastoma to demonstrate that supernatant of IL-2 and anti-CD16 antibodies treated NK cells as well as IFN-γ and TNF-α mediate differentiation of cancer stem-like tumors 35,36. Therefore, the purpose of treating lung cancer stem-like cells hA549 with the supernatants of IL-2 and anti-CD16 antibodies NK cells in the current study was to determine if IL-2 and anti-CD16 antibodies treated NK cells can mediate differentiation of lung cancer stem-like cells as well. Similar to our previous observation in several other cancer types differentiated lung cancer cell line H292 was found to be more sensitive to chemotherapeutic drugs compared to hA549 22. Similar to differentiated lung cancer cell line H292, NK cell-induced differentiated hA549 tumors were found to exhibit reduced sensitivity to NK cell-induced killing but were highly susceptible to chemo-drug-induced killing compared to hA549 (Fig. 2).
